# Deep Lattice Networks and Partial Monotonic Functions

**Seungil You, David Ding, Kevin Canini, Jan Pfeifer, Maya R. Gupta**
Google Research
1600 Amphitheatre Parkway, Mountain View, CA 94043
{siyou,dwding,canini,janpf,mayagupta}@google.com

## Abstract

We propose learning deep models that are monotonic with respect to a user-specified set of inputs by alternating layers of linear embeddings, ensembles of lattices, and calibrators (piecewise linear functions), with appropriate constraints for monotonicity, and jointly training the resulting network. We implement the layers and projections with new computational graph nodes in TensorFlow and use the Adam optimizer and batched stochastic gradients. Experiments on benchmark and real-world datasets show that six-layer monotonic deep lattice networks achieve state-of-the art performance for classification and regression with monotonicity guarantees.

## 1 Introduction

We propose building models with multiple layers of lattices, which we refer to as *deep lattice networks* (DLNs). While we hypothesize that DLNs may generally be useful, we focus on the challenge of learning flexible partially-monotonic functions, that is, models that are guaranteed to be monotonic with respect to a user-specified subset of the inputs. For example, if one is predicting whether to give someone else a loan, we expect and would like to constrain the prediction to be monotonically increasing with respect to the applicant's income, if all other features are unchanged. Imposing monotonicity acts as a regularizer, improves generalization to test data, and makes the end-to-end model more interpretable, debuggable, and trustworthy.

To learn more flexible partial monotonic functions, we propose architectures that alternate three kinds of layers: linear embeddings, calibrators, and ensembles of lattices, each of which is trained discriminatively to optimize a structural risk objective and obey any given monotonicity constraints. See Fig. 2 for an example DLN with nine such layers.

Lattices are interpolated look-up tables, as shown in Fig. 1. Lattices have been shown to be an efficient nonlinear function class that can be constrained to be monotonic by adding appropriate sparse linear inequalities on the parameters [1], and can be trained in a standard empirical risk minimization framework [2, 1]. Recent work showed lattices could be jointly trained as an ensemble to learn flexible monotonic functions for an arbitrary number of inputs [3].

Calibrators are one-dimensional lattices, which nonlinearly transform a single input [1]; see Fig. 1 for an example. They have been used to pre-process inputs in two-layer models: calibrators-then-linear models [4], calibrators-then-lattice models [1], and calibrators-then-ensemble-of-lattices model [3]. Here, we extend their use to discriminatively normalize between other layers of the deep model, as well as act as a pre-processing layer. We also find that using a calibrator for a last layer can help nonlinearly transform the outputs to better match the labels.

We first describe the proposed DLN layers in detail in Section 2. In Section 3, we review more related work in learning flexible partial monotonic functions. We provide theoretical results characterizing the flexibility of the DLN in Section 4, followed by details on our open-source TensorFlow imple-

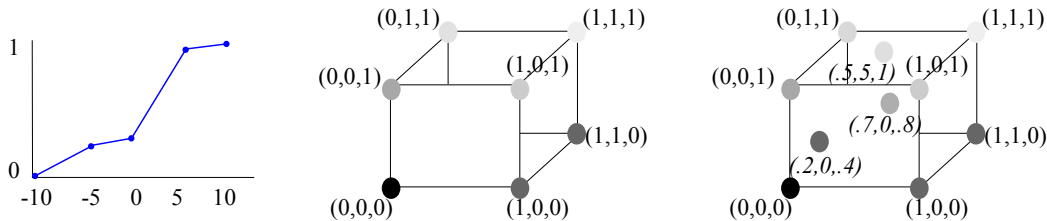

Figure 1: **Left:** Example calibrator (1-d lattice) with fixed input range $[-10, 10]$ and five fixed uniformly-spaced keypoints and corresponding discriminatively-trained outputs (look-up table values values). **Middle:** Example lattice on three inputs in fixed input range $[0, 1]^3$, with 8 discriminatively-trained parameters (shown as gray-values), each corresponding to one of the $2^3$ vertices of the unit hypercube. The parameters are linearly interpolated for any input $[0, 1]^3$ to form the lattice function's output. If the parameters are increasing in any direction, then the function is monotonic increasing in that direction. In this example, the gray-value parameters get lighter in all three directions, so the function is monotonic increasing in all three inputs. **Right:** Three examples of lattice values are shown in italics, each interpolated from the 8 lattice parameters.

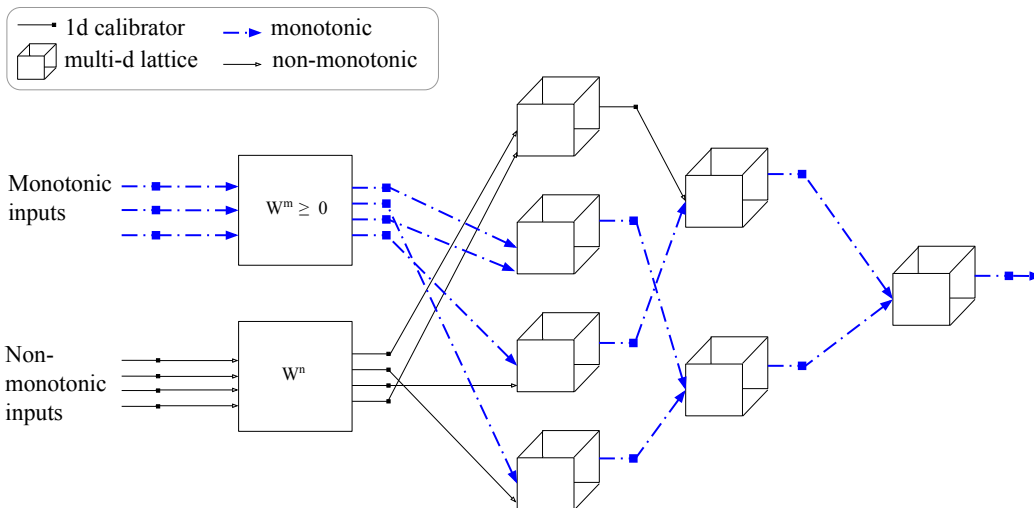

Figure 2: Illustration of a nine-layer DLN: calibrators, linear embedding, calibrators, ensemble of lattices, calibrators, ensemble of lattices, calibrators, lattice, calibrator.

mentation and numerical optimization choices in Section 5. Experimental results demonstrate the potential on benchmark and real-world scenarios in Section 6.

## 2 Deep Lattice Network Layers

We describe in detail the three types of layers we propose for learning flexible functions that can be constrained to be monotonic with respect to any subset of the inputs. Without loss of generality, we assume monotonic means monotonic non-decreasing (one can flip the sign of an input if non-increasing monotonicity is desired). Let $x_t \in \mathbb{R}^{D_t}$ be the input vector to the $t$th layer, with $D_t$ inputs, and let $x_t[d]$ denote the $d$th input for $d = 1, \ldots, D_t$. Table 1 summarizes the parameters and hyperparameters for each layer. For notational simplicity, in some places we drop the notation $t$ if it is clear in the context. We also denote as $x_t^m$ the subset of $x_t$ that are to be monotonically constrained, and as $x_t^n$ the subset of $x_t$ that are non-monotonic.

**Linear Embedding Layer:** Each linear embedding layer consists of two linear matrices, one matrix $W_t^m \in \mathbb{R}^{D_{t+1}^m \times D_t^m}$ that linearly embeds the monotonic inputs $x_t^m$, and a separate matrix $W_t^n \in \mathbb{R}^{(D_{t+1} - D_{t+1}^m) \times (D_t - D_t^m)}$ that linearly embeds non-monotonic inputs $x_t^n$, and one bias vector

$b_t$. To preserve monotonicity on the embedded vector $W_t^m x_t^m$, we impose the following linear inequality constraints:

$$W_t^m[i,j] \geq 0 \text{ for all } (i,j). \tag{1}$$

The output of the linear embedding layer is:

$$x_{t+1} \quad = \quad \begin{bmatrix} x_{t+1}^m \\ x_{t+1}^n \end{bmatrix} = \begin{bmatrix} W_t^m x_t^m \\ W_t^n x_t^n \end{bmatrix} + b_t$$

Only the first $D_{t+1}^m$ coordinates of $x_{t+1}$ needs to be a *monotonic* input to the $t+1$ layer. These two linear embedding matrices and bias vector are discriminatively trained.

**Calibration Layer:** Each calibration layer consists of a separate one-dimensional piecewise linear transform for each input at that layer, $c_{t,d}(x_t[d])$ that maps $\mathbb{R}$ to $[0,1]$, so that

$$x_{t+1} := [c_{t,1}(x_t[1]) \quad c_{t,2}(x_t[2]) \quad \cdots \quad c_{t,D_t}(x_t[D_t])]^T .$$

Here each $c_{t,d}$ is a 1D lattice with $K$ key-value pairs $(a \in \mathbb{R}^K, b \in \mathbb{R}^K)$, and the function for each input is linearly interpolated between the two $b$ values corresponding to the input's surrounding $a$ values. An example is shown on the left in Fig. 1.

Each 1D calibration function is equivalent to a sum of weighted-and-shifted Rectified linear units (ReLU), that is, a calibrator function $c(x[d]; a, b)$ can be equivalently expressed as

$$c(x[d]; a, b) = \sum_{k=1}^{K} \alpha[k]\text{ReLU}(x - a[k]) + b[1], \tag{2}$$

where

$$\alpha[k] \quad := \quad \begin{cases} \frac{b[k+1]-b[k]}{a[k+1]-a[k]} - \frac{b[k]-b[k-1]}{a[k]-a[k-1]} & \text{for } k = 2, \cdots, K-1 \\ \frac{b[2]-b[1]}{a[2]-a[1]} & \text{for } k = 1 \\ -\frac{b[K]-b[K-1]}{a[K]-a[K-1]} & \text{for } k = K \end{cases}$$

However, enforcing monotonicity and boundedness constraints for the calibrator output is much simpler with the $(a, b)$ parameterization of each keypoint's input-output values, as we discuss shortly.

Before training the DLN, we fix the input range for each calibrator to $[a_{\min}, a_{\max}]$, and we fix the $K$ keypoints $a \in \mathbb{R}^K$ to be uniformly-spaced over $[a_{\min}, a_{\max}]$. Inputs that fall outside $[a_{\min}, a_{\max}]$ are clipped to that range. The calibrator output parameters $b \in [0, 1]^K$ are discriminatively trained.

For monotonic inputs, we can constrain the calibrator functions to be monotonic by constraining the calibrator parameters $b \in [0, 1]^K$ to be monotonic, by adding the linear inequality constraints

$$b[k] \leq b[k+1] \text{ for } k = 1, \ldots, K-1 \tag{3}$$

into the training objective [3]. We also experimented with constraining all calibrators to be monotonic (even for non-monotonic inputs) for more stable/regularized training.

**Ensemble of Lattices Layer:** Each ensemble of lattices layer consists of $G$ lattices. Each lattice is a linearly interpolated multidimensional look-up table; for an example, see the middle and right pictures in Fig. 1. Each $S$-dimensional look-up table takes inputs over the $S$-dimensional unit hypercube $[0, 1]^S$, and has $2^S$ parameters $\theta \in \mathbb{R}^{2^S}$, specifying the lattice's output for each of the $2^S$ vertices of the unit hypercube. Inputs in-between the vertices are linearly interpolated, which forms a smooth but nonlinear function over the unit hypercube. Two interpolation methods have been used, multilinear interpolation and simplex interpolation [1] (also known as Lovász extension [5]). We use multilinear interpolation for all our experiments, which can be expressed $\psi(x)^T \theta$ where the non-linear feature transformation $\psi(x) : [0, 1]^S \to [0, 1]^{2^S}$ are the $2^S$ linear interpolation weights that input $x$ puts on each of the $2^S$ parameters $\theta$ such that the interpolated value for $x$ is $\psi(x)^T \theta$, and $\psi(x)[j] = \Pi_{d=1}^S x[d]^{v_j[d]}(1-x[d])^{1-v_j[d]}$, where $v_j[\cdot] \in 0, 1$ is the coordinate vector of the $j$th vertex of the unit hypercube, and $j = 1, \cdots, 2^D$. For example, when $S = 2$, $v_1 = (0, 0), v_2 = (0, 1), v_3 = (1, 0), v_4 = (1, 1)$ and $\psi(x) = ((1 - x[1])(1 - x[2]), (1 - x[1])x[2], x[1](1 - x[2]), x[1]x[2])$.

The ensemble of lattices layer produces $G$ outputs, one per lattice. When initializing the DLN, if the $t+1$th layer is an ensemble of lattices, we randomly permute the outputs of the previous layer

Table 1: DLN layers and hyperparameters

| Layer $t$ | Parameters | Hyperparameters |
|---|---|---|
| Linear Embedding | $b_t \in \mathbb{R}^{D_{t+1}}, W_t^m \in \mathbb{R}^{D_{t+1}^m \times D_t^m}$, $W_t^n \in \mathbb{R}^{(D_{t+1}-D_{t+1}^m) \times (D_t - D_t^m)}$ | $D_{t+1}$ |
| Calibrators | $B_t \in \mathbb{R}^{D_t \times K}$ | $K \in \mathbb{N}^+$ keypoints, input range $[\ell, u]$ |
| Lattice Ensemble | $\theta_{t,g} \in \mathbb{R}^{2^{S_t}}$ for $g = 1, \ldots, G_t$ | $G_t$ lattices $S_t$ inputs per lattice |

to be assigned to the $G_{t+1} \times S_{t+1}$ inputs of the ensemble. If a lattice has at least one monotonic input, then that lattice's output is constrained to be a monotonic input to the next layer to guarantee end-to-end monotonicity. Each lattice is constrained to be monotonic by enforcing monotonicity constraints on each pair of lattice parameters that are adjacent in the monotonic directions; for details see Gupta et al. [1].

**End-to-end monotonicity:** The DLN is constructed to preserve end-to-end monotonicity with respect to a user-specified subset of the inputs. As we described, the parameters for each component (matrix, calibrator, lattice) can be constrained to be monotonic with respect to a subset of inputs by satisfying certain linear inequality constraints [1]. Also if a component has a monotonic input, then the output of that component is treated as a monotonic input to the following layer. Because the composition of monotonic functions is monotonic, the constructed DLN belongs to the partial monotonic function class. The arrows in Figure 2 illustrate this construction, *i.e.*, how the $t$th layer output becomes a monotonic input to $t + 1$th layer.

## 2.1 Hyperparameters

We detail the hyperparameters for each type of DLN layer in Table 1. Some of these hyperparameters constrain each other since the number of outputs from each layer must be equal to the number of inputs to the next layer; for example, if you have a linear embedding layer with $D_{t+1} = 1000$ outputs, then there are 1000 inputs to the next layer, and if that next layer is a lattice ensemble, its hyperparameters must obey $G_t \times S_t = 1000$.

## 3 Related Work

Low-dimensional monotonic models have a long history in statistics, where they are called shape constraints, and often use isotonic regression [6]. Learning monotonic single-layer neural nets by constraining the neural net weights to be positive dates back to Archer and Wang in 1993 [7], and that basic idea has been re-visited by others [8, 9, 10, 11], but with some negative results about the obtainable flexibility, even with multiple hidden layers [12]. Sill [13] proposed a three-layer monotonic network that used monotonic linear embedding and max-and-min-pooling. Daniels and Velikova [12] extended Sill's result to learn a partial monotonic function by combining min-max-pooling, also known as adaptive logic networks [14], with partial monotonic linear embedding, and showed that their proposed architecture is a universal approximator for partial monotone functions. None of these prior neural networks were demonstrated on problems with more than $D = 10$ features, nor trained on more than a few thousand examples. For our experiments we implemented a positive neural network and a min-max-pooling network [12] with TensorFlow.

This paper extends recent work in learning multidimensional flexible partial monotonic 2-layer networks consisting of a layer of calibrators followed by an ensemble of lattices [3], with parameters appropriately constrained for monotonicity, which built on earlier work of Gupta et al. [1]. This work differs in three key regards.

First, we alternate layers to form a deeper, and hence potentially more flexible, network. Second, a key question addressed in Canini et al. [3] is how to decide which features should be put together in each lattice in their ensemble. They found that random assignment worked well, but required large ensembles. They showed that smaller (and hence faster) models with the same accuracy could be

trained by using a heuristic pre-processing step they proposed (*crystals*) to identify which features interact nonlinearly. This pre-processing step requires training a lattice for each pair of inputs to judge that pair's strength of interaction, which scales as $O(D^2)$, and we found it can be a large fraction of overall training time for $D > 50$.

We solve this problem of determining which inputs should interact in each lattice by using a linear embedding layer before an ensemble of lattices layer to discriminatively and adaptively learn during training how to map the features to the first ensemble-layer lattices' inputs. This strategy also means each input to a lattice can be a linear combination of the features. This use of a jointly trained linear embedding is the second key difference to that prior work [3].

The third difference is that in previous work [4, 1, 3], the calibrator keypoint values were fixed a priori based on the quantiles of the features, which is challenging to do for the calibration layers mid-DLN, because the quantiles of their inputs are evolving during training. Instead, we fix the keypoint values uniformly over the bounded calibrator domain.

## 4 Function Class of Deep Lattice Networks

We offer some results and hypotheses about the function class of deep lattice networks, depending on whether the lattices are interpolated with multilinear interpolation (which forms multilinear polynomials), or simplex interpolation (which forms locally linear surfaces).

### 4.1 Cascaded multilinear lookup tables

We show that a deep lattice network made up only of cascaded layers of lattices (without intervening layers of calibrators or linear embeddings) is equivalent to a single lattice defined on the $D$ input features if multilinear interpolation is used. It is easy to construct counter-examples showing that this result does *not* hold for simplex-interpolated lattices.

**Lemma 1.** *Suppose that a lattice has $L$ inputs that can each be expressed in the form $\theta_i^T \psi(x[s_i])$, where the $s_i$ are mutually disjoint and $\psi$ represents multilinear interpolation weights. Then the output can be expressed in the form $\hat{\theta}^T \hat{\psi}(x[\cup s_i])$. That is, the lattice preserves the functional form of its inputs, changing only the values of the coefficients $\theta$ and the linear interpolation weights $\psi$.*

*Proof.* Each input $i$ of the lattice can be expressed in the following form:

$$f_i = \theta_i^T \psi(x[s_i]) = \sum_{k=1}^{2^{|s_i|}} \theta_i[v_{ik}] \prod_{d \in s_i} x[d]^{v_{ik}[d]} (1 - x[d])^{1 - v_{ik}[d]}$$

This is a multilinear polynomial on $x[s_i]$. The output can be expressed in the following form:

$$F = \sum_{j=1}^{2^L} \theta_i[v_j] \prod_{i=1}^{L} f_i^{v_j[i]} (1 - f_i)^{1 - v_j[i]}$$

Note the product in the expression: $f_i$ and $1 - f_i$ are both multilinear polynomials, but within each term of the product, only one is present, since one of the two has exponent $0$ and the other has exponent $1$. Furthermore, since each $f_i$ is a function of a different subset of $x$, we conclude that the entire product is a multilinear polynomial. Since the sum of multilinear polynomials is still a multilinear polynomial, we conclude that $F$ is a multilinear polynomial. Any multilinear polynomial on $k$ variables can be converted to a $k$-dimensional multilinear lookup table, which concludes the proof. $\square$

Lemma 1 can be applied inductively to every layer of cascaded lattices down to the final output $F(x)$. We have shown that cascaded lattices using multilinear interpolation is equivalent to a single multilinear lattice defined on all $D$ features.

### 4.2 Universal approximation of partial monotone functions

Theorem 4.1 in [12] states that partial monotone linear embedding followed by min and max pooling can approximate any partial monotone functions on the hypercube up to arbitrary precision given

sufficiently high embedding dimension. We show in the next lemma that simplex-interpolated lattices can represent min or max pooling. Thus one can use a DLN constructed with a linear embedding layer followed by two cascaded simplex-interpolated lattice layers to approximate any partial monotone function on the hypercube.

**Lemma 2.** *Let $\theta_{\min} = (0, 0, \cdots, 0, 1) \in \mathbb{R}^{2^n}$ and $\theta_{\max} = (1, 0, \cdots, 0) \in \mathbb{R}^{2^n}$, and $\psi_{simplex}$ be the simplex interpolation weights. Then*

$$\begin{aligned} \min(x[0], x[1], \cdots, x[n]) &= \psi_{simplex}(x)^T \theta_{\min} \\ \max(x[0], x[1], \cdots, x[n]) &= \psi_{simplex}(x)^T \theta_{\max} \end{aligned}$$

*Proof.* From the definition of simplex interpolation [1], $\psi_{simplex}(x)^T \theta = \theta[1]x[\pi[1]] + \cdots + \theta[2^n]x[\pi[n]]$, where $\pi$ is the sorted order such that $x[\pi[1]] \geq \cdots \geq x[\pi[n]]$, and due to sparsity, $\theta_{\min}$ and $\theta_{\max}$ selects the min and the max. □

### 4.3 Locally linear functions

If simplex interpolation [1] (aka the Lovász extension) is used, the deep lattice network produces a locally linear function, because each layer is locally linear, and compositions of locally linear functions are locally linear. Note that a $D$ input lattice interpolated with simplex interpolation has $D!$ linear pieces [1]. If one cascades an ensemble of $D$ lattices into a lattice, then the number of possible locally linear pieces is of the order $O((D!)!)$.

## 5 Numerical Optimization Details for the DLN

**Operators:** We implemented 1D calibrators and multilinear interpolation over a lattice as new C++ operators in TensorFlow [15] and express each layer as a computational graph node using these new and existing TensorFlow operators. Our implementation is open sourced and can be found in `https://github.com/tensorflow/lattice`. We use the Adam optimizer [16] and batched stochastic gradients to update model parameters. After each batched gradient update, we project parameters to satisfy their monotonicity constraints. The linear embedding layer's constraints are element-wise non-negativity constraints, so its projection clips each negative component to zero. This projection can be done in $O(\text{\# of elements in a monotonic linear embedding matrix})$. Projection for each calibrator is isotonic regression with chain ordering, which we implement with the pool-adjacent-violator algorithm [17] for each calibrator. This can be done in $O(\text{\# of calibration keypoints})$. Projection for each lattice is isotonic regression with partial ordering that imposes $O(S2^S)$ linear constraints for each lattice [1]. We solved it with consensus optimization and alternating direction method of multipliers [18] to parallelize the projection computations with a convergence criterion of $\epsilon = 10^{-7}$. This can be done in $O(S2^S \log(1/\epsilon))$.

**Initialization:** For linear embedding layers, we initialize each component in the linear embedding matrix with IID Gaussian noise $\mathcal{N}(2, 1)$. The initial mean of 2 is to bias the initial parameters to be positive so that they are not clipped to zero by the first monotonicity projection. However, because the calibration layer before the linear embedding outputs in $[0, 1]$ and thus is expected to have output $\mathbb{E}[x_t] = 0.5$, initializing the linear embedding with a mean of 2 introduces an initial bias: $\mathbb{E}[x_{t+1}] = \mathbb{E}[W_t x_t] = D_t$. To counteract that we initialize each component of the bias vector, $b_t$, to $-D_t$, so that the initial expected output of the linear layer is $\mathbb{E}[x_{t+1}] = \mathbb{E}[W_t x_t + b_t] = 0$.

We initialize each lattice's parameters to be a linear function spanning $[0, 1]$, and add IID Gaussian noise $\mathcal{N}(0, \frac{1}{S^2})$ to each parameter, where $S$ is the number of input to a lattice. We initialize each calibrator to be a linear function that maps $[x_{\min}, x_{\max}]$ to $[0, 1]$ (and did not add any noise).

## 6 Experiments

We present results on the same benchmark dataset (Adult) with the same monotonic features as in Canini et al. [3], and for three problems from Google where the monotonicity constraints were specified by product groups. For each experiment, every model considered is trained with monotonicity guarantees on the same set of inputs. See Table 2 for a summary of the datasets.

Table 2: Dataset Summary

| Dataset | Type | # Features (# Monotonic) | # Training | # Validation | # Test |
|---|---|---|---|---|---|
| Adult | Classify | 90 (4) | 26,065 | 6,496 | 16,281 |
| User Intent | Classify | 49 (19) | 241,325 | 60,412 | 176,792 |
| Rater Score | Regress | 10 (10) | 1,565,468 | 195,530 | 195,748 |
| Usefulness | Classify | 9 (9) | 62,220 | 7,764 | 7,919 |

Table 3: User Intent Case Study Results

| | Validation Accuracy | Test Accuracy | # Parameters | $G \times S$ |
|---|---|---|---|---|
| DLN | 74.39% | 72.48% | 27,903 | $30 \times 5D$ |
| Crystals | 74.24% | 72.01% | 15,840 | $80 \times 7D$ |
| Min-Max network | 73.89% | 72.02% | 31,500 | $90 \times 7D$ |

For classification problems, we used logistic loss, and for the regression, we used squared error. For each problem, we used a validation set to optimize the hyperparameters for each model architecture: the learning rate, the number of training steps, etc. For an ensemble of lattices, we tune the number of lattices, $G$, and number of inputs to each lattice, $S$. All calibrators for all models used a fixed number of 100 keypoints, and set $[-100, 100]$ as an input range.

In all experiments, we use the six-layer DLN architecture: Calibrators $\rightarrow$ Linear Embedding $\rightarrow$ Calibrators $\rightarrow$ Ensemble of Lattices $\rightarrow$ Calibrators $\rightarrow$ Linear Embedding, and validate the number of lattices in the ensemble $G$, number of inputs to each lattice, $S$, the Adam stepsize and number of loops.

For crystals [3] we validated the number of ensembles, $G$, and number of inputs to each lattice, $S$, as well as Adam stepsize and number of loops. For min-max net [12], we validated the number of groups, $G$, and dimension of each group $S$, as well as Adam stepsize and number of loops.

For datasets where all features are monotonic, we also train a deep neural network with a non-negative weight matrix and ReLU as an activation unit with a final fully connected layer with non-negative weight matrix, which we call *monotonic DNN*, akin to the proposals of [7, 8, 9, 10, 11]. We tune the depth of hidden layers, $G$, and the activation units in each layer $S$.

All the result tables are sorted by their validation accuracy, and contain an additional column for chosen hyperparameters; $2 \times 5D$ means $G = 2$ and $S = 5$.

## 6.1 User Intent Case Study (Classification)

For this real-world Google problem, the problem is to classify the user intent. This experiment is set-up to test generalization ability to non-IID test data. The train and validation examples are collected from the U.S., and the test set is collected from 20 other countries, and as a result of this difference between the train/validation and test distributions, there is a notable difference between the validation and the test accuracy. The results in Table 3 show a $0.5\%$ gain in test accuracy for the DLN.

## 6.2 Adult Benchmark Dataset (Classification)

We compare accuracy on the benchmark Adult dataset [19], where a model predicts whether a person's income is at least $50,000 or not. Following Canini et al. [1], we require all models to be monotonically increasing in capital-gain, weekly hours of work and education level, and the gender wage gap. We used one-hot encoding for the other categorical features, for 90 features in total. We randomly split the usual train set [19] 80-20 and trained over the $80\%$, and validated over the $20\%$.

Table 4: Adult Results

|  | Validation Accuracy | Test Accuracy | # Parameters | $G \times S$ |
|---|---|---|---|---|
| DLN | 86.50% | 86.08% | 40,549 | $70 \times 5D$ |
| Crystals | 86.02% | 85.87% | 3,360 | $60 \times 4D$ |
| Min-Max network | 85.28% | 84.63% | 57,330 | $70 \times 9D$ |

Results in Table 4 show the DLN provides better validation and test accuracy than the min-max network or crystals.

### 6.3 Rater Score Prediction Case Study (Regression)

For this real-world Google problem, we train a model to predict a rater score for a candidate result, where each rater score is averaged over 1-5 raters, and takes on 5-25 possible real values. All 10 monotonic features are required to be monotonic. Results in Table 5 show the DLN has very test MSE than the two-layer crystals model, and much better MSE than the other monotonic networks.

Table 5: Rater Score Prediction (Monotonic Features Only) Results

|  | Validation MSE | Test MSE | # Parameters | $G \times S$ |
|---|---|---|---|---|
| DLN | 1.2078 | 1.2096 | 81,601 | $50 \times 9D$ |
| Crystals | 1.2101 | 1.2109 | 1,980 | $10 \times 7D$ |
| Min-Max network | 1.3474 | 1.3447 | 5,500 | $100 \times 5D$ |
| Monotonic DNN | 1.3920 | 1.3939 | 2,341 | $20 \times 100D$ |

### 6.4 Usefulness Case Study (Classifier)

For this real-world Google problem, we train a model to predict whether a candidate result adds useful information given the presence of another result. All 9 features are required to be monotonic. Table 6 shows the DLN has slightly better validation and test accuracy than crystals, and both are notably better than the min-max network or positive-weight DNN.

Table 6: Usefulness Results

|  | Validation Accuracy | Test Accuracy | # Parameters | $G \times S$ |
|---|---|---|---|---|
| DLN | 66.08% | 65.26% | 81,051 | $50 \times 9D$ |
| Crystals | 65.45% | 65.13% | 9,920 | $80 \times 6D$ |
| Min-Max network | 64.62% | 63.65% | 4,200 | $70 \times 6D$ |
| Monotonic DNN | 64.27% | 62.88% | 2,012 | $1 \times 1000D$ |

## 7 Conclusions

In this paper, we proposed combining three types of layers, (1) calibrators, (2) linear embeddings, and (3) multidimensional lattices, to produce a new class of models we call *deep lattice networks* that combines the flexibility of deep networks with the regularization, interpretability and debuggability advantages that come with being able to impose monotonicity constraints on some inputs.

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
