[Supplementary Material]

# Deep Lattice Networks and Partial Monotonic Functions : Supplementary Material

**Seungil You, David Ding, Kevin Canini, Jan Pfeifer, Maya Gupta**
Google Inc.
1600 Amphitheatre Parkway, Mountain View, CA 94043
{siyou,dwding,canini,janpf,mayagupta}@google.com

# 1 Additional results from experiments

## 1.1 User intent dataset

Figure 1: Validation accuracy vs. Test accuracy on the user intent dataset. Each model architecture was trained using a range of hyperparameters, and DLN shows better performance over other architectures.

| Model | $S$ | $G$ | # of steps | batch size | learning rate | Training accuracy | Validation accuracy | Test accuracy |
|---|---|---|---|---|---|---|---|---|
| DLN | 5 | 30 | 160000 | 1000 | 0.01 | 75.71% | 74.39% | 72.48% |
| Crystals | . | . | . | . | 0.001 | . | 74.24% | 72.01% |
| DLN | 5 | 40 | 80000 | 1000 | 0.01 | 76.04% | 74.21% | 71.42% |
| DLN | 5 | 30 | 80000 | 1000 | 0.01 | 75.33% | 74.15% | 71.59% |
| DLN | 5 | 40 | 160000 | 1000 | 0.01 | 76.35% | 74.00% | 71.77% |
| DLN | 7 | 30 | 80000 | 1000 | 0.01 | 76.17% | 73.97% | 71.34% |
| DLN | 7 | 40 | 160000 | 1000 | 0.01 | 76.99% | 73.94% | 71.33% |
| DLN | 7 | 30 | 160000 | 1000 | 0.01 | 76.43% | 73.93% | 71.51% |
| DLN | 5 | 40 | 160000 | 1000 | 0.001 | 75.76% | 73.90% | 71.62% |
| Min-Max | 7 | 90 | 40000 | 1000 | 0.01 | 74.19% | 73.89% | 72.02% |
| Min-Max | 9 | 80 | 24000 | 1000 | 0.01 | 74.12% | 73.89% | 72.21% |
| DLN | 7 | 30 | 160000 | 1000 | 0.0001 | 74.63% | 73.83% | 71.91% |
| Min-Max | 6 | 80 | 40000 | 1000 | 0.01 | 74.25% | 73.81% | 71.92% |
| DLN | 7 | 40 | 80000 | 1000 | 0.01 | 76.33% | 73.80% | 71.67% |
| DLN | 5 | 40 | 80000 | 1000 | 0.001 | 75.19% | 73.80% | 71.23% |
| DLN | 5 | 30 | 80000 | 1000 | 0.001 | 74.96% | 73.80% | 72.01% |
| Min-Max | 8 | 90 | 40000 | 1000 | 0.01 | 74.02% | 73.79% | 72.14% |
| Min-Max | 8 | 100 | 24000 | 1000 | 0.01 | 74.04% | 73.79% | 72.10% |
| Min-Max | 8 | 70 | 32000 | 1000 | 0.01 | 74.16% | 73.75% | 72.05% |
| DLN | 5 | 40 | 160000 | 1000 | 0.0001 | 74.48% | 73.74% | 71.92% |

Table 1: Performance over user intent dataset with top 20 hyperparameter configurations including model type, ranked by validation accuracy.

## 1.2    Adult dataset

Figure 2: Validation accuracy vs. Test accuracy on the Adult dataset. Each model architecture was trained using a range of hyperparameters, and DLN shows better performance over other architectures.

| Model | $S$ | $G$ | # of steps | batch size | learning rate | Training accuracy | Validation accuracy | Test accuracy |
|---|---|---|---|---|---|---|---|---|
| DLN | 5 | 70 | 16000 | 1000 | 0.0001 | 87.11% | 86.50% | 86.08% |
| DLN | 9 | 50 | 16000 | 1000 | 0.0001 | 87.12% | 86.45% | 86.06% |
| DLN | 9 | 60 | 16000 | 1000 | 0.0001 | 87.19% | 86.45% | 86.04% |
| DLN | 9 | 70 | 16000 | 1000 | 0.0001 | 87.34% | 86.35% | 86.01% |
| DLN | 5 | 60 | 16000 | 1000 | 0.0001 | 87.06% | 86.22% | 86.01% |
| DLN | 5 | 50 | 16000 | 1000 | 0.0001 | 87.06% | 86.19% | 86.14% |
| DLN | 5 | 70 | 8000 | 1000 | 0.0001 | 86.48% | 86.07% | 85.62% |
| DLN | 5 | 50 | 8000 | 1000 | 0.0001 | 86.32% | 86.04% | 85.65% |
| Crystals | 4 | 60 | 5000 | 100 | 0.01 | 86.03% | 86.02% | 85.87% |
| DLN | 5 | 60 | 8000 | 1000 | 0.0001 | 86.30% | 86.01% | 85.72% |
| DLN | 9 | 60 | 8000 | 1000 | 0.1 | 85.94% | 85.95% | 85.88% |
| DLN | 9 | 50 | 8000 | 1000 | 0.0001 | 86.47% | 85.88% | 85.59% |
| DLN | 9 | 70 | 8000 | 1000 | 0.0001 | 86.45% | 85.88% | 85.62% |
| DLN | 9 | 60 | 8000 | 1000 | 0.0001 | 86.31% | 85.84% | 85.55% |
| DLN | 9 | 60 | 8000 | 1000 | 0.001 | 90.16% | 85.53% | 85.01% |
| DLN | 5 | 50 | 8000 | 1000 | 0.001 | 89.85% | 85.42% | 85.22% |
| DLN | 5 | 50 | 8000 | 1000 | 0.1 | 85.24% | 85.31% | 85.07% |
| Min-Max | 9 | 70 | 8000 | 1000 | 0.1 | 87.06% | 85.28% | 84.63% |
| DLN | 5 | 60 | 8000 | 1000 | 0.001 | 90.06% | 85.27% | 84.85% |
| DLN | 9 | 50 | 8000 | 1000 | 0.001 | 90.11% | 85.11% | 84.76% |

Table 2: Performance over the Adult dataset with top 20 hyperparameter configurations including model type, ranked by validation accuracy.

## 1.3 Rater score dataset

Figure 3: Validation mean squared error vs. Test mean squared error on the rater score dataset. Each model architecture was trained using a range of hyperparameters, and DLN shows better performance over other architectures.

| Model | $S$ | $G$ | # of epochs | batch size | learning rate | Training MSE | Validation MSE | Test MSE |
|---|---|---|---|---|---|---|---|---|
| DLN | 9 | 50 | 80 | 1000 | 0.01 | 1.2100 | 1.2078 | 1.2096 |
| DLN | 5 | 40 | 80 | 1000 | 0.01 | 1.2112 | 1.2088 | 1.2103 |
| Crystals | 10 | 7 | 2500 | 100 | 0.0001 | . | 1.2101 | 1.2109 |
| DLN | 9 | 50 | 40 | 1000 | 0.01 | 1.2138 | 1.2117 | 1.2130 |
| DLN | 5 | 50 | 80 | 1000 | 0.01 | 1.2143 | 1.2125 | 1.2135 |
| DLN | 9 | 40 | 80 | 1000 | 0.01 | 1.2174 | 1.2145 | 1.2167 |
| DLN | 9 | 50 | 40 | 1000 | 0.001 | 1.2203 | 1.2171 | 1.2187 |
| DLN | 9 | 50 | 40 | 1000 | 0.001 | 1.2203 | 1.2171 | 1.2187 |
| DLN | 9 | 30 | 80 | 1000 | 0.001 | 1.2209 | 1.2181 | 1.2196 |
| DLN | 5 | 30 | 80 | 1000 | 0.01 | 1.2215 | 1.2185 | 1.2201 |
| DLN | 5 | 30 | 80 | 1000 | 0.01 | 1.2215 | 1.2185 | 1.2201 |
| DLN | 9 | 30 | 40 | 1000 | 0.01 | 1.2237 | 1.2205 | 1.2226 |
| DLN | 9 | 30 | 40 | 1000 | 0.01 | 1.2237 | 1.2205 | 1.2226 |
| DLN | 5 | 30 | 40 | 1000 | 0.01 | 1.2235 | 1.2205 | 1.2216 |
| DLN | 5 | 30 | 40 | 1000 | 0.01 | 1.2235 | 1.2205 | 1.2216 |
| DLN | 9 | 50 | 80 | 1000 | 0.001 | 1.2216 | 1.2210 | 1.2204 |
| DLN | 5 | 40 | 40 | 1000 | 0.001 | 1.2254 | 1.2232 | 1.2239 |
| DLN | 5 | 40 | 40 | 1000 | 0.001 | 1.2254 | 1.2232 | 1.2239 |
| DLN | 9 | 40 | 80 | 1000 | 0.001 | 1.2243 | 1.2238 | 1.2224 |
| DLN | 5 | 40 | 80 | 1000 | 0.001 | 1.2252 | 1.2239 | 1.2239 |

Table 3: Performance over the rater score dataset with top 20 hyperparameter configurations including model type, ranked by validation mean squared error.

## 1.4 Usefulness dataset

Figure 4: Validation accuracy vs. Test accuracy on the usefulness dataset. Each model architecture was trained using a range of hyperparameters, and DLN shows better performance over other architectures.

## 2 Timing results

This section contains the total training wall time for the best deep lattice network model and Crystals for each dataset. For deep lattice network training, we used a virtual machine with 40 CPU cores and used TensorFlow to train a model, but did not compile the code with vector instructions. For Crystals, we used 3.5GHz Intel Ivy Bridge processor and used highly optimized C++ implementations to train a model. Because of this difference, the timing results here are not comparable to each other, and there are many rooms for improvements in the implementation, for example utilizing GPGPU instructions. So this result should not be interpreted as a proper timing benchmark study. However we report this number for the readers who might be curious about the training wall time in each experiment. See Table 5.

| Model | $S$ | $G$ | # of epochs | batch size | learning rate | Training accuracy | Validation accuracy | Test accuracy |
|---|---|---|---|---|---|---|---|---|
| DLN | 9 | 50 | 400 | 1000 | 0.01 | 66.18% | 66.08% | 65.26% |
| DLN | 5 | 30 | 400 | 1000 | 0.01 | 66.01% | 65.78% | 65.55% |
| DLN | 9 | 50 | 200 | 1000 | 0.01 | 65.47% | 65.73% | 64.80% |
| DLN | 7 | 50 | 400 | 1000 | 0.001 | 64.79% | 65.71% | 63.90% |
| DLN | 5 | 30 | 200 | 1000 | 0.01 | 65.31% | 65.57% | 64.53% |
| DLN | 7 | 40 | 400 | 1000 | 0.01 | 65.83% | 65.55% | 65.42% |
| DLN | 5 | 50 | 400 | 1000 | 0.001 | 64.84% | 65.49% | 63.87% |
| DLN | 7 | 30 | 200 | 1000 | 0.01 | 64.90% | 65.49% | 64.69% |
| Crystals | 80 | 6 | 2500 | 100 | 0.0027 | . | 65.45% | 65.13% |
| DLN | 5 | 40 | 400 | 1000 | 0.001 | 64.63% | 65.43% | 63.61% |
| DLN | 7 | 30 | 400 | 1000 | 0.001 | 64.93% | 65.43% | 63.91% |
| DLN | 9 | 30 | 200 | 1000 | 0.01 | 65.18% | 65.37% | 64.13% |
| DLN | 9 | 40 | 200 | 1000 | 0.01 | 65.17% | 65.37% | 64.78% |
| DLN | 7 | 40 | 400 | 1000 | 0.001 | 64.89% | 65.34% | 64.19% |
| DLN | 9 | 40 | 400 | 1000 | 0.001 | 64.92% | 65.30% | 63.65% |
| DLN | 7 | 30 | 200 | 1000 | 0.1 | 64.28% | 65.27% | 63.91% |
| DLN | 7 | 30 | 400 | 1000 | 0.1 | 64.93% | 65.27% | 64.51% |
| DLN | 5 | 30 | 400 | 1000 | 0.001 | 64.66% | 65.26% | 63.56% |
| DLN | 5 | 40 | 400 | 1000 | 0.01 | 65.52% | 65.21% | 64.63% |
| DLN | 5 | 40 | 200 | 1000 | 0.01 | 64.87% | 65.14% | 64.29% |

Table 4: Performance over the usefulness dataset with top 20 hyperparameter configurations including model type, ranked by validation accuracy.

| Dataset | Model | Total wall time (hh:mm:ss) | # of parameters | Batch size | $G$ | $S$ | Avg. throughput (batch per sec) |
|---|---|---|---|---|---|---|---|
| User intent | DLN | 06:36:36 | 27,903 | 1,000 | 30 | 5 | 6.594801713 |
| User intent | Crystals | 09:25:44 | 15,840 | 100 | 80 | 7 | . |
| Adult | DLN | 01:38:28 | 40,549 | 1,000 | 70 | 5 | 2.779140688 |
| Adult | Crystals | 00:28:56 | 3,360 | 100 | 60 | 4 | . |
| Rater score | DLN | 28:29:01 | 81,601 | 1,000 | 50 | 9 | 1.192121216 |
| Rater score | Crystals | 09:49:02 | 1,980 | 100 | 10 | 7 | . |
| Usefulness | DLN | 07:07:18 | 81,051 | 1,000 | 50 | 9 | 0.9728394583 |
| Usefulness | Crystals | 17:50:21 | 9,920 | 100 | 80 | 6 | . |

Table 5: Total training wall time in each experiment.