[Reviews · NeurIPS 2017]

Reviewer 1



The authors propose an network architecture with a combination of linear encoding, lattices (non-linear function) and piece-wise linear functions with constraints of monotonicity to preserve partial monotonicity w.r.t user specified subset of input variables and the target. Questions: Excuse me for being unfamiliar with corresponding literature. Why is it required to specify the feature subset on which we want to impose monotonicity w.r.t the prediction? cant it be figured by the neural network from the data? (especially when training on large number of samples) The authors chose a positive DNN for comparison. Assuming the monotonicity constraints are resulting in regularization in DLNs (resulting in better test scores), the authors can consider comparing against a regular neural network with out any constraints. How the authors chose the architecture of the DNN for each dataset?

Reviewer 2



This paper proposes a deep neural network structure which acts as a function approximator for partial monotonic functions. The previous works that try to approximate a partial monotonic function are only one/two layers deep with hand designed architectures. The key contribution of this paper is to develop a deep version of partial monotonic function. A deep network composed with the components proposed in this paper is proved to have end to end monotonicity property. While I'm not very familiar with this line of work, my feeling is that deep neural net with functional class constraint is a useful topic. From the previous work listed in this paper, I feel that this work is the first to design an end to end trainable deep network that is proved to be monotonic.

Reviewer 3



This paper proposes a novel deep model that is monotonic to user specified inputs, called deep lattice networks (DLN's), which is composed of layers of linear embeddings, ensembles of lattices and calibrators. The results about the function classes of DLN's are also provided. Authors should put more effort to make the paper readable for the people that are not familiar with the technicality. However the contributions are interesting and significant. As [3] proposed a 2 layer model, which has a layer of calibrators followed by an ensemble of lattices, the advantages of proposed deeper model could be presented in a more convincing way by showing better results as the results seem closer to Crystals and it is not clear whether the produced results are stable. The computational complexity should be also discussed clearly. Each projection step in optimization should be explained in more detail. What is the computational complexity of the projection for the calibrator? What proportion of training time the projections take? Are the projections accurate or suboptimal in the first iterations? And how the overall training time compares to baselines? Does the training of the proposed model produces bad local minima for deeper models? At most how many layers could be trained with current optimization procedure? It seems most of the results are based on the data that is not publicly available. To make the results reproducible, authors should add more publicly available data to experiments. Also artificial data would help to demonstrate some sanity checks, e.g. a case that two layer [3] did not work well but deeper model work better, the intermediate layer visualizations etc . Authors should also explain better why the model has interpretability and debuggability advantages.